molecular biology/physiology

transcriptome analyses, midgut, hindgut, unpredictable chronic stress, Atlantic salmon, parr

**Author for correspondence:**
Signe Dille Løvmo
e-mail: signe.d.lovmo@ntnu.no

# Mid and hindgut transcriptome profiling analysis of Atlantic salmon (*Salmon salar*) under unpredictable chronic stress

Signe Dille Løvmo[2], Angelico Madaro[1], Paul Whatmore[1], Tora Bardal[2], Mari-Ann Ostensen[2], Simen R. Sandve[3] and Rolf Erik Olsen[1,2]

[1]Institute of Marine Research, Animal Welfare Science Group, Matredal 5984, Norway
[2]Department of Biology, Norwegian University of Science and Technology, Trondheim 7491, Norway
[3]Norwegian University of Life Sciences, Ås, Norway

SDL, 0000-0003-1894-3149

The intestinal epithelium is a selectively permeable barrier for nutrients, electrolytes and water, while maintaining effective protection against pathogens. Combinations of stressors throughout an animal's life, especially in agriculture and aquaculture settings, may affect the regular operativity of this organ with negative consequences for animal welfare. In the current study, we report the effects of a three-week unpredictable chronic stress (UCS) period on the intestinal morphology and transcriptome response of Atlantic salmon (*Salmon salar*) parr midgut and hindgut. Midgut and hindgut from both control and UCS fish were collected for histology and RNA-sequencing analysis to identify respective changes in the membrane structures and putative genes and pathways responding to UCS. Histological analysis did not show any significant effect on morphometric parameters. In the midgut, 1030 genes were differentially expressed following UCS, resulting in 279 genes which were involved in 13 metabolic pathways, including tissue repair pathways. In the hindgut, following UCS, 591 differentially expressed genes were detected with 426 downregulated and 165 upregulated. A total of 53 genes were related to three pathways. Downregulated genes include cellular senescence pathways, p53 signalling and cytokine–cytokine receptor pathways. The overall results corroborate that salmon parr were at least partly habituating to the UCS treatment. In midgut, the main upregulation was related to cell growth and repair, while in the hindgut there

were indications of the activated apoptotic pathway, reduced cell repair and inhibited immune/anti-inflammatory capacity. This may be the trade-off between habituating to UCS and health resilience. This study suggests possible integrated genetic regulatory mechanisms that are tuned when farmed Atlantic salmon parr attempt to cope with UCS.

## 1. Introduction

The gut barrier has several significant biological functions and in addition to the primary function of absorbing nutrients, it is also an essential barrier between the fish and the external environment, controlling the loss of nutrients and preventing the uptake of noxious substances. Furthermore, it also harbours an extensive microbiota that can aid in nutrient utilization and protect against pathogen agents [1]. However, several factors can interfere with gut functionality, with serious implications for the health and welfare of the organism [2].

Because of the recognized and considerable impact to human and high-value livestock health, stress-related alteration of gastrointestinal integrity has received considerable attention in mammalian and human systems [3]. Emotional and physical acute and chronic stress has been shown to reduce the physical barrier function leading to increased intestinal permeability, functional dyspepsia, irritable bowel syndrome and peptic ulcer disease [4–8].

Fish experience stress under both wild and culture conditions, and there is an increasing body of information on the effect of stressors regarding intensity, duration, predictability and controllability [9]. Stress causes barrier dysfunction and increased intestinal permeability [10,11] and shortening of intestinal villi structures and inflammation [12–14]. In European eel (*Anguilla anguilla* L.), social chronic stress has been shown to cause stomach epithelium atrophy, gastric glands degeneration in addition to swollen mitochondria and impaired cell-to-cell contacts [15]. In addition, in carp (*Cyprinus carpio*), catch and transportation dependent stress caused loss of intestinal mucus-producing cells (goblet cells) and detachment of columnar absorptive epithelial cells [16]. However, quite often results are unclear and even inconsistent. While high stocking densities have been reported to reduce goblet cells (GC) in number and size, as well as villi length in channel catfish (*Ictalurus punctatus*) [17], no such effects were seen in rainbow trout (*Oncorhynchus mykiss*) [18]. These differences are likely a result of species differences and the lack of proper comparative experimental protocols. Furthermore, there is limited information on the transcriptional mechanisms activated in fish intestine during stress, though some studies have investigated the transcriptional profile of the fish intestine after stress [19]. In Asian Seabass (*Lates calcarifer*), partial repression of the intestinal immune system is seen after being challenged by pathogen or immune modulators [20]. Sending Japanese Medaka to the international space station resulted in a greater transcriptional change in the gut compared to other organs, while almost no difference was seen in morphology [21].

Most stress-related experiments on fish have examined acute stress, or a chronic exposure to a design of repeated single stressors [22–26]. Although these studies have provided important information on basal fish physiology, they would not be good models for studying chronic stress, because fish tend to adapt to most normal stressors within a week or so [27–29]. The same had been observed earlier in mammalian studies [30–33], underlining the importance of developing new tools that could study the mechanisms of true chronic stress. To study these mechanisms the unpredictable chronic stress (UCS) paradigm was devised, where an animal would be exposed to several mild stressors in a random, unpredictable manner [34]. As a consequence, the animal would develop typical chronic stress conditions without being exposed to severe physical treatment [34,35].

The UCS paradigm has been more recently adopted in fish studies, with considerable focus on developing experimental methods that would not damage or harm the fish [36]. A recent study by Madaro *et al.* [37] used a three-week UCS paradigm on Atlantic salmon (*Salmon salar*) parr focusing on the regulation of the hypothalamic-pituitary-interrenal (HPI) stress axis. The procedure caused no physical harm to the fish, but a decrease in appetite and malregulation of the HPI axis suggested the procedure had been successful. By the end of the three weeks, some habituation was observed, and fish appetite started to increase.

The aim of this study was to investigate the effect of UCS on Atlantic salmon smolt intestine. We report the effects of a three-week UCS trial on the intestinal histology and transcriptome response of Atlantic salmon parr midgut and hindgut. The overall results corroborate that salmon parr were at least partly habituating to the UCS treatment, activating an adaptive mechanism to ensure essential

**Table 1.** Description of the stressors randomly given to Atlantic salmon parr throughout the experiment [37].

| stressor | time (min) | description |
|---|---|---|
| chasing | 5 | stirring in the tank with a net |
| netting | 3 | net and release fish with a dip net including brief air exposure (±1 s) |
| temperature shock 12–4℃ | 120 | reduction of the water temperature from 12℃ to 4℃ and up to 12℃ |
| temperature shock 12–19℃ | 120 | rise of the water temperature from 12℃ to 19℃ and down to 12℃ |
| noise | 5 | knocking on the tank with a metal object |
| darkness + flashlight | 5 | turn off the tank lighting and use of a white intermittent LED light |
| brief hypoxia | 5 | closure of water inflow until the oxygen saturation of the water reaches 40% |
| emptying the tank | 5 | removal of the tank plug while leaving the water flow open with a constant 3 cm deep layer of water as a result |

functions relating to cell/tissue integrity while the fish immunity decreased. Results provide new insight into the understanding of integrated genetic adaptive pathways activated by farmed Atlantic salmon parr intestine in order to cope with UCS.

# 2. Material and methods

## 2.1. Experimental animals and facilities

Atlantic salmon eggs (Aqua Gen strain, Aqua Gen AS, Trondheim, Norway) were hatched and reared at the Institute of Marine Research, Matre, Norway. Experimental fish were kept in 10 000 l outdoor tanks under natural conditions (9°C). A month before the start of the experiment, 740 fish (approx. 10 months old) were transferred into six indoor tanks (400 l; density: 7 kg fish/tank) supplied with flow-through freshwater. Fish were kept at 12°C on a 12 : 12 photoperiod with a water flow of 15 l min$^{-1}$, which provided an approximately 92% oxygen saturation. Fish were fed with dry pellets (2 mm Skretting Nutra Olympic, Stavanger, Norway) until satiety three times a day with automatic feeders. (Arvo-tec feeding units: Arvo-Tec T drum 2000, Huutokoski, Finland). Uneaten feed was collected from the outlet pipes. Tank conditions were monitored and regulated by a fully automated system (SD Matre, Normatic AS, Nordfjordeid, Norway).

## 2.2. Experimental design

The experimental design is described in Madaro *et al.* [37]. Briefly, at the beginning of the experiment (4 February 2013), six tanks were divided into two groups ($n = 3$ replicates), of which one received a set of random stressors (UCS) while the other was left undisturbed (control group). The UCS group was stressed three times per day (08.30 h, 13.00 h and 17.00 h) using a total of eight types of stressors given in random and unpredictable order throughout one week, and this protocol was then repeated for the next two weeks over a total of 23 days. Stressors (table 1) were chosen such that there would be no physical damage to the fish (such as major scale loss or fin damages). Disturbance for the control group was reduced to a minimum and limited to routine practices of tank maintenance and sampling. Fish were fed with dry feed for a duration of 1 h, 30–60 min after each stress event (i.e. at 09.00–10.00 h, 13.30–14.30 h and 17.30–18.30 h). All uneaten food was collected, and dry weight recorded 15 min after feeding. On day 23, fish were sampled from each tank, subjected to UCS and compared with fish from each of the unstressed control tanks.

## 2.3. Sampling

The sampling is described in Madaro *et al.* [37]. Briefly, fish were starved for 12 h before sampling. All groups were sampled early in the morning (09.00) and all samples required less than 2 min per fish to

be processed. Fish received an overdose of anaesthesia (100 mg $1^{-1}$ Tricaine methanesulfonate, buffered with 100 mg $l^{-1}$ sodium bicarbonate (Finquel®vet.)) which rendered them completely motionless (no opercular movement) within 10 s of immersion. For each individual fish, fork length and body mass were recorded, and blood samples collected using a 1 ml heparinized syringe with a 23 G needle. Blood samples were centrifuged at 13 000 rpm for 2 min and stored at −80°C for cortisol analysis.

Tissue samples from midgut and hindgut for the transcriptome study were collected from all individuals, rinsed in PBS and stored in RNAlater (RNAlater® RNA Stabilization Solution, Life Technology, Oslo, Norway) at 4°C for 24 h before transferring to −80°C until RNA isolation. From the same fish, a fragment of midgut and hindgut was collected, rinsed in PBS and fixed in Karnovsky fixative for histological analyses. At the end of the trial, no fish were observed with any physical damage and there was no mortality due to the UCS regime.

## 2.4. Blood analyses

The concentration of plasma cortisol was quantified by radioimmunoassay (Perkin Elmer, Groningen, The Netherlands), and evaluated using two-way ANOVA followed by Fisher's LSD test, as previously described in Madaro *et al.* [37].

## 2.5. Histological analyses

Midgut and hindgut samples were taken from five unstressed fish and six UCS-exposed fish ($N = 5/6$, from all three replicates), and embedded in Technovit® 7100 (Kulzer, Germany). Two micrometre sections were obtained by a Microtome Jung Autocut 2055 (Leica, Germany) using carbide metal blade (Leica TC65 microtome blade). Sections were stained with HE (haematoxylin and eosin) (Merck, Germany), mounted with Neo-Mount® (Merck, Germany) and scanned at 40× magnification using a digital slide scanner (NanoZoomer SQ, Hamamatsu Photonics, Japan) using the image software NDP.view2 (Hamamatsu Photonics, Japan). All measurements were done manually on the scanned images. Within each preparation, four to five villi structures were randomly selected, and villi height calculated, all with a longitudinal cross section through the lamina propria of the fold. Within the marked area, GC, intraepithelial leucocytes and granulocytes within the mucosa were counted. In addition, the degree of widening of each villi lamina propria was measured at apical, mid and basal part. Halfway up the villi, total enterocyte height was measured. The parameters were as follows: (i) the abundance of GC within the villi, numbers per 0.1 $\text{mm}^{-2}$; (ii) intraepithelial leucocytes, numbers per 0.1 $\text{mm}^{-2}$; (iii) number of granulocytes in the lamina propria, numbers per 0.1 $\text{mm}^{-2}$; (iv) the degree of widening of the lamina propria at apical, mid and basal regions. Data reported as the mean of 5 measurements per area; (v) enterocyte height at midsections of the villi (mean of 10 measurements) and (vi) villi height (mean of 6 per preparation). For each measurement, the mathematical mean for each fish was calculated and used as the basis for statistical analysis giving $N = 5$ for control and $N = 6$ for UCS. Stress and control groups were compared using student's *t*-test with significance accepted as $p < 0.05$.

## 2.6. RNA extraction

RNA extractions were carried out at NTNU Institute for Biology, Trondheim, Norway. Midgut and hindgut total RNA were isolated using RNeasy Plus Universal Mini Kit (Qiagen, Hilden Germany) according to the manufacturer's instructions. RNA concentration and purity were determined using Nanodrop 8000 (Thermo Scientific, Wilmington, USA). RNA integrity was checked by using Agilent 2100 Bioanalyzer (Agilent Technologies, Santa Clara, CA, USA). A RIN equal or above eight confirmed excellent quality RNA. Three fish from the control group and four fish from the UCS group were selected from two and three replicates, respectively, and samples from midgut and hindgut were used to construct the sequencing libraries (14 samples in total).

## 2.7. Library preparation

Library preparations were carried out at the Centre for Integrative Genetics (CIGENE, ÅS, Norway) using TruSeq Stranded mRNA Sample Prep HS Protocol (Illumina, San Diego, CA, USA), selecting for 500 bp fragments. Briefly, polyA containing mRNA molecules were purified from the total RNA according to the polyA selection method and then fragmented with the fragmentation buffer. Cleaved RNA fragments

were primed with random hexamers into first-strand cDNA using reverse transcriptase and random primers. After, a double-stranded cDNA is synthesized by replacing the RNA template with a second cDNA strand. The synthesized cDNA was then subjected to end-repair, phosphorylation and 'A' base addition, according to Illumina's library construction protocol. Libraries were sequenced using single-end high-throughput mRNA sequencing (RNA-Seq) on Illumina Hiseq 2500 (Illumina, San Diego, CA, USA) at the Norwegian Sequencing Centre (Oslo, Norway).

## 2.8. RNA-seq raw data quality control and mapping

Across all 14 samples, a total of 354.7 million single-end, 120 bp reads were sequenced, with an average 17.7 million reads per sample.

Sequences were quality trimmed using cutadapt (v. 1.8.1), whereupon adapter sequences, low-quality bases (Phred score < 20) and reads with fewer than 40 bases were removed. Trimmed reads were aligned to the salmon reference genome (ICSASG_v2) using STAR (v. 2.5.2a). Read counts per gene were quantified from aligned reads using HTSeq-count (v. 0.6.1p1), with gene names annotated from the NCBI salmon genome annotation (available for download at https://v1.salmobase.org/download.html). Raw sequence files (fastq) were uploaded to the NCBI sequence read archive (SRA). SRA accession number is SRP169832.

The variance of gene expression between and within experimental groups was examined using hierarchical clustering. Plots were generated using the tables of normalized read counts per gene. Hierarchical clustering and heat maps were generated using the R package heatmap, using a between-sample euclidean distance matrix generated by the base R package 'dist'.

## 2.9. Differential expression analysis and functional annotations

Downstream analysis including differential expression (DE) and functional annotation was completed in R v. 3.4.1 (http://cran.rproject.org/). Genes were initially annotated with Entrez gene identifiers and were subsequently annotated to gene symbols and gene descriptions. While gene symbols are presented in this paper, Entrez IDs were used for KEGG and GO enrichment analysis. Electronic supplementary tables of DE genes and enriched pathways contain all three identifiers for each associated gene—Entrez ID, gene symbol and gene description. To quantify levels of DE per gene, a table of read counts per gene per sample were input into DESeq2 [38], which estimates significant DE based on a Wald test, using a negative binomial generalized linear model. DESeq2 performs internal normalization for both sequencing depth (library size) and RNA composition, based on the geometric mean per gene across all treatment samples.

Significantly DE genes were considered those that had a false-discovery adjusted (Benjamini–Hochberg) $p$-value of less than 0.05. We did not include a fold change cutoff but relied on DESeq2 to identify which genes were significantly DE, as DESeq2 controls for false positives and is sensitive to small, true differences.

Functional annotation for KEGG and GO pathways was completed using the ClusterProfiler package v. 3.6.0 [39]. The statistical machinery for enrichment analysis in ClusterProfiler is provided by the DOSE (disease ontology semantic and enrichment analysis) package v. 3.4.0 [40]. Using DOSE, we identified enriched pathways by performing an over-representation test, based on a hypergeometric model. KEGG pathways and GO terms were determined to be significantly enriched if they had Benjamini–Hochberg adjusted $p$-values ($q$-values) of less than 0.05.

# 3. Results

## 3.1. Feeding, growth and plasma cortisol

Food consumption, growth data and plasma cortisol were analysed for the whole trial and are presented in Madaro *et al.* [37] (electronic supplementary material, figures S13–S15). The UCS group showed a reduced appetite throughout the experiment; however, the appetite increased in the last 7 days. The UCS group also showed a significant reduced growth in terms of body mass compared to the control group after 23 days. Plasma cortisol were significantly elevated in the UCS fish compared to control fish throughout the experiment, though a lower response was observed during the acute stress test (5 min chasing) at the end of the experiment (results presented in Madaro *et al.* [37]).

## 3.2. Histological evaluation

The sum of results is seen in table 2 and histological sections are represented in figure 1. The villi in the midgut were in the range 300–407 µm in the hindgut, but with relatively large variation. Enterocyte at the midsection of the villi had a mean total height of around 50 µm, generally slightly larger in the hindgut section. The lamina propria was generally widest at the apical part of the villi in midgut followed by the basal region; however, there was no significant different in size. In hindgut, differences were less notable, although apical parts still were generally wider than other parts of the villi. The number of GC was around 60–70 per 0.1 mm$^{-2}$ in midgut, which was roughly double what was found in the hindgut sections. Intraepithelial leucocytes were scattered throughout the basal parts of the enterocytes and totalled around 70 0.1 mm$^{-2}$. Granulocytes in the lamina propria were difficult to identify with HE staining and were only positively identified in the midgut where they totalled around 6 per 0.1 mm$^{-2}$. Subjecting the fish to three weeks of chronic stress did not appear to have any effect on the gross histology as measured by the current parameters (figure 1).

## 3.3. Differentially expressed genes

Gene identification was based on the Atlantic salmon ICSASG_v2 reference genome. Annotation to the 44 868 Atlantic salmon reference genes produced around 29% of duplicated Entrez gene identifiers, but the vast majority (greater than 99%) of these duplicates were transfer RNA. None of the DE genes were duplicates. Of the 1030 genes that were differentially expressed (DE) in the midgut, 329 were downregulated and 701 upregulated (figure 2a; electronic supplementary material, table S4). In the hindgut 591 genes were differentially expressed, of which 426 were downregulated and 165 upregulated (figure 2b; electronic supplementary material, table S3). There were 114 genes that were concordantly differentially expressed in both midgut and hindgut (electronic supplementary material, table S6). The top 2 concordant genes, both strongly downregulated, were rho-associated protein kinase 2-like (−6.10 log2fc in hindgut and −5.32 log2fc in midgut) and von Willebrand factor A domain-containing protein 7-like (−4.43 and −5.03 log2fc in hindgut and midgut, respectively).

In terms of global expression patterns, hierarchical clustering showed that midgut and hindgut samples clustered separately, except for one midgut sample which grouped with the hindgut samples. UCS and control treatment groups showed less separation in both midgut and hindgut, indicating less overall within-group variation (UCS versus control) than between the gut regions (midgut versus hindgut) (electronic supplementary material, figure S1).

## 3.4. Significantly changed pathway analysis

In the midgut 13 KEGG pathways were significantly enriched (Benjamini–Hochberg adjusted $p$-values < 0.05), with most pathways involved in cell metabolism and DNA replication and repair (figure 3a, table 3). Within the two most significantly changed pathways, ribosome ($n = 49$, adjusted $p = 3.8 \times 10^{-21}$) and cell cycle (figure 4, $n = 37$, adjusted $p = 1.2 \times 10^{-10}$), all associated genes in these pathways were upregulated.

In the hindgut three pathways were significantly enriched (figure 3b, table 4). In the cytokine–cytokine receptor pathway ($n = 25$, adjusted $p = 2.3 \times 10^{-06}$) all DE genes were downregulated, whereas cellular senescence ($n = 18$, adjusted $p = 5.1 \times 10^{-03}$) and p53 signalling pathway ($n = 10$, adjusted $p = 4.0 \times 10^{-03}$) contained both down and upregulated genes. Figures for all significantly enriched KEGG pathways, showing up and downregulated genes, can be found in the electronic supplementary material, figure S2–S13.

Note that tables 3 and 4 show the Atlantic salmon gene symbol for each DE gene per enriched pathway. As the Atlantic salmon reference genome has been constructed relatively recently, most of these are 'placeholder' symbols in the form of 'LOC….' and are associated with homologous genes in other species. More information for each of these gene symbols can be found in electronic supplementary material, tables S1 and S2, including Entrez ID, full gene description, log2 fold change and false-discovery adjusted $p$-values.

## 4. Discussion

The current study presents a snapshot of the transcriptome response of Atlantic salmon parr midgut and hindgut after three weeks of a UCS regime and displays mechanisms that may be tuned in order to cope

**Table 2.** Histological evaluation of salmon parr intestine before and after stress. Data are means ± s.d. of five control and six UCS histological preparations. Significance $p < 0.05$ between control and stressed groups are written in the table. Cont = control group, UCS = unpredictable chronic stressed group, l_leucoc = intraepithelial leucocytes, Granuloc = granulocytes, — = could not be measured accurately on the preparations.

| | lamina propria width (µm) | | | cells (0.1 mm$^{-2}$) | | | height (µm) | |
|---|---|---|---|---|---|---|---|---|
| | basal | mid | apical | goblet | l_leucoc | granuloc | enterocyte | villi |
| **midgut** | | | | | | | | |
| cont | 12.01 ± 3.12 | 8.36 ± 1.91 | 14.52 ± 5.24 | 58.8 ± 22.7 | 63.6 ± 10.8 | 5.19 ± 1.00 | 51.6 ± 7.5 | 343.5 ± 64.3 |
| UCS | 8.31 ± 0.25 | 7.33 ± 1.47 | 14.25 ± 2.07 | 71.1 ± 11.2 | 76.8 ± 6.5 | 6.21 ± 0.96 | 48.3 ± 1.8 | 300.1 ± 15.4 |
| | $p = 0.052$ | $p = 0.554$ | $p = 0.356$ | $p = 0.939$ | $p = 0.497$ | $p = 0.797$ | $p = 0.266$ | $p = 0.794$ |
| **hindgut** | | | | | | | | |
| cont | 8.54 ± 1.99 | 7.77 ± 1.43 | 10.44 ± 2.97 | 32.8 ± 4.3 | 73.1 ± 10.1 | — | 54.8 ± 1.7 | 375.1 ± 75.3 |
| UCS | 7.25 ± 132 | 7.59 ± 1.36 | 7.76 ± 1.68 | 26.5 ± 4.3 | 69.1 ± 10.0 | — | 57.3 ± 3.2 | 406.9 ± 36.0 |
| | $p = 0.408$ | $p = 0.308$ | $p = 0.154$ | $p = 0.723$ | $p = 0.739$ | | $p = 0.338$ | $p = 0.226$ |

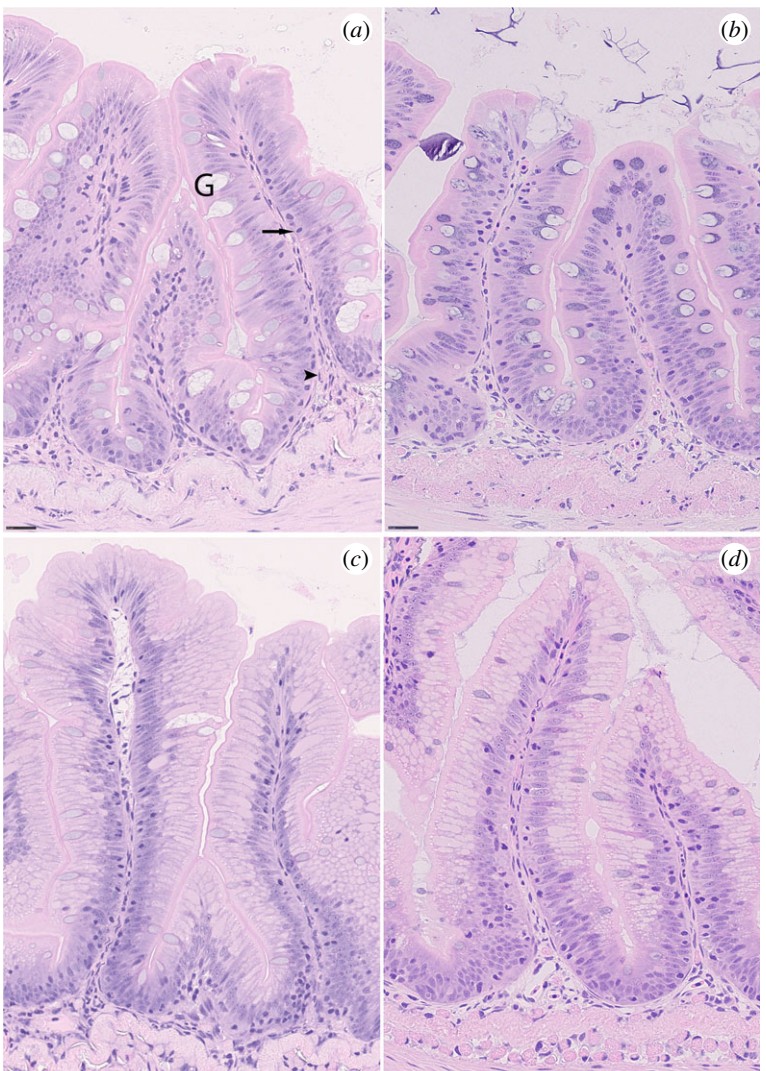

**Figure 1.** Midgut sections in Atlantic salmon parr before (*a*) and after three weeks (*b*) exposure to unpredictable stress. Hindgut sections from the same fish before (*c*) and after three weeks after stress (*d*). G = goblet cells, arrow = Intraepithelial lymphocytes, arrow head = Granulocytes. Bar = 25 um.

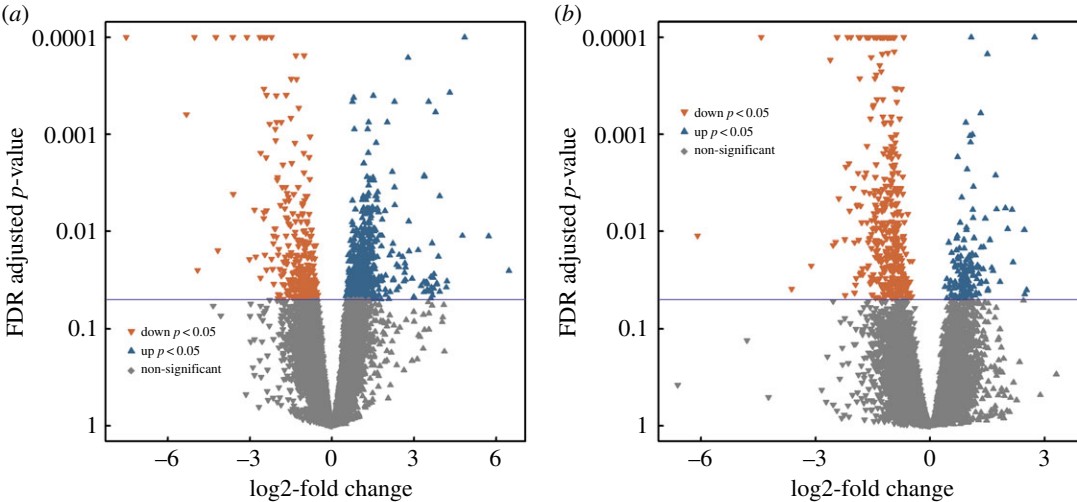

**Figure 2.** Overview of fold changes in gene expression in Atlantic salmon midgut (*a*) and hindgut (*b*). *y*-axis indicates the statistical adjusted *p*-value, and *x*-axis indicates the log2-fold change between reads per gene. Up arrows indicate upregulated and down arrows represent downregulated genes. Significantly DE genes are coloured (Benjamini–Hochberg adjusted *p*-values< = 0.05) and non-significant DE genes are grey.

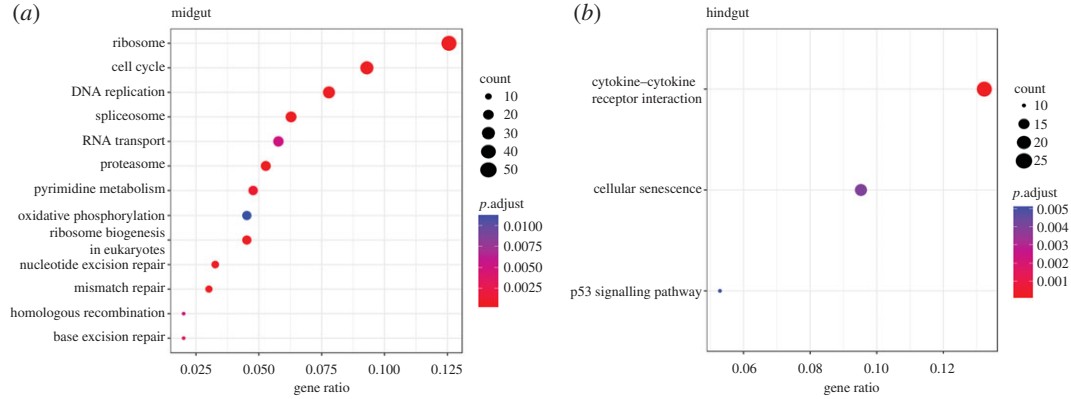

**Figure 3.** Summary of the significantly enriched KEGG pathways in the midgut (*a*) and the hindgut (*b*). *y*-axes denote specific pathways, and *x*-axes indicate the enrichment ratio. The size of the dots represents the number of the significantly expressed genes involved into each corresponding pathway, while colour of the dots shows the false-discovery adjusted *p*-value of the pathways.

with long-term exposure to stress. The midgut transcriptome data clearly reveals upregulation of several processes that sustain the renewal of the tissue following chronic stress at different levels, from DNA replication to protein synthesis. Conversely, in the hindgut UCS resulted in mostly downregulation of transcriptome regulatory pathways involved in the immune-inflammatory host response, cell ageing and apoptosis, in the attempt to cope with chronic stress and limit the '*wear and tear*' of the body [41,42]. A comparison of both histological sections for midgut and hindgut did not show significant differences in their morphology, number of GC, intraepithelial leucocytes and granulocytes.

In humans, stress is linked to several gastrointestinal diseases, like peptic ulcer, functional dyspepsia, inflammatory bowel disease and irritable bowel syndrome [5]. Stress has also been shown to effect the gastrointestinal tract across fish species. For example, former studies on fish have reported that acute stress caused detachment of the intestinal mucus layer, increased permeability and loss of junctional complexes [10,11]. However, most of these studies are limited to acute stress and single stressor, and an increasing number of studies are also showing that many fish, including salmonids, have a remarkable ability to adapt to stressors [29,37,43].

Chronic stress is also known to reduce appetite and growth in salmonids [37,44,45]. Accordingly, at the end of the 23-day experiment, the stressed group showed a null growth, but no loss of body weight compared the control fish. Initially, fish had a reduced feed intake and elevated cortisol response following a stress test, showing all signs of chronic stress [37]. However, appetite increased in the UCS group while the cortisol response to stress decreased towards the end of the trial, clearly showing signs of habituation to the treatment. The hypothesis that fish started to habituate to stress may also be supported by the fact that after the end of the trial, following smoltification and seawater transfer, the UCS group showed an overall higher specific growth rate compared with unstressed control groups (data showed in Vindas *et al.* [46]). The recovery of the fishes' appetite and habituation to stressors in the final part of the experiment may also explain the lack of histological features expected after stress, such as epithelial flattening or changes in immune cell density [12–14]. Sampling at an earlier timepoint may have revealed different morphological features. Transcriptomic data, on the other hand, indicate some reduction in expression of the *mucin-2* gene in both midgut and hindgut, but the reduction was not related to changes in GC size or abundance. Transcriptomic data also did not show any significant effects on tight junction or cell-to-cell adherence which is a common feature in chronic stressed animals [12,14]. The lack of appetite in the initial part of the trail could also influence the intestinal morphology. However, a previous study done on fasting Atlantic salmon showed that the intestine was regenerated to a normal state after only 7 days of refeeding [47].

The expression profile in the midgut sections showed that the main effect of UCS was the upregulation of genes related to regeneration pathways such as regulation of cell cycle and DNA replication pathways, indicating that the fish are using more energy to maintain homeostasis in the midgut, and therefore are still affected by the UCS even after habituation. The enriched KEGG pathways include cell cycle, Ribosome and DNA replication among others (figure 3*a*; electronic supplementary material, figures S2, S4–S7, S9–S13). Of interest was that UCS caused upregulation of all the genes that translate for the mini-chromosome maintenance protein complex (MCM, figure 4), which is essential for cell growth, initiation and elongation steps in DNA replication [48].

**Table 3.** List of enriched (false-discovery adjusted $p < 0.05$) KEGG pathways containing significantly differentially expressed genes (adjusted $p = <0.05$) in the Atlantic salmon midgut.

| description | count | $p$-value | adjusted $p$ | $q$-value | genes |
|---|---|---|---|---|---|
| ribosome | 49 | $6.3 \times 10^{-23}$ | $3.8 \times 10^{-21}$ | $3.3 \times 10^{-21}$ | LOC106585178, rl23a, LOC106592737, rl37, LOC106588488, mrps14, LOC106588994, rps20, LOC106570097, LOC106596289, LOC106567051, LOC106563677, LOC106572990, LOC106603708, LOC106582975, LOC106568161, LOC106606582, LOC106570028, LOC106605701, LOC106612239, LOC106586974, rpl9, rpl18, LOC106604773, rs30, mrps11, LOC106587490, mrps6, LOC106607032, LOC106561938, LOC106568981, rl6, rl3, rs17, rps7, LOC106577194, LOC106577194, LOC106568029, LOC106604455, fau, LOC106587585, LOC106610183, mrpl30, LOC106611516, mrpl9, rl37a, rps17, LOC106581113, LOC106563517, LOC106583574 |
| cell cycle | 37 | $3.0 \times 10^{-12}$ | $1.2 \times 10^{-10}$ | $1.1 \times 10^{-10}$ | ccne2, LOC106607826, LOC106566901, LOC106580445, LOC106607055, LOC106607002, mcm3, ccnb1, ccnb2, cdc20, LOC106573204, LOC106585071, espl1, cdc2, ccna2, LOC106611545, mcm7, LOC106582108, plk1, LOC106566708, LOC106601305, LOC106569474, mcm2, LOC106583830, cdc7, orc5, mcm5, LOC106568170, LOC106569128, LOC106574843, cdc45, skp1, LOC106577502, LOC100306849, mc6zb, LOC106583719, LOC106570699 |
| DNA replication | 30 | $1.5 \times 10^{-28}$ | $1.8 \times 10^{-26}$ | $1.6 \times 10^{-26}$ | LOC106580445, LOC106607055, rfc5, prim2, mcm3, dna2, LOC106585071, rpa3, pole3, rfc2, LOC106606382, mcm7, LOC106564474, LOC106606474, LOC106569474, LOC106612933, mcm2, prim1, lig1, mcm5, LOC106569128, LOC106605109, rfc4, pole, fen1, pola2, mc6zb, LOC106583719, pole2 |
| spliceosome | 25 | $9.8 \times 10^{-06}$ | $1.5 \times 10^{-04}$ | $1.3 \times 10^{-04}$ | LOC106609479, LOC106576293, rsmb, LOC106588760, LOC106561682, sf3a2, LOC106572841, LOC106588262, LOC106581107, LOC106588762, LOC106585463, LOC106612789, LOC106607257, LOC106600990, hnrpg, sfrs1, ppih, LOC106565620, roa1, LOC106611702, LOC106581844, LOC106611629, LOC106563120, LOC106570608, LOC106603799 |
| RNA transport | 23 | $7.6 \times 10^{-04}$ | $8.3 \times 10^{-03}$ | $7.4 \times 10^{-03}$ | LOC106605059, 4ebp, LOC100019680, eif4e1a, nup93, LOC106574637, rgp1, LOC106567676, if2a, LOC106607164, LOC106607749, eif3 g, rpp30, nup188, LOC106577251, LOC106610986, LOC106607209, LOC106581844, LOC106577302, if4e, LOC106571672, ran1, LOC106563585 |

(Continued.)

**Table 3.** (*Continued.*)

| description | count | p-value | adjusted p | q-value | genes |
|---|---|---|---|---|---|
| proteasome | 21 | $8.4 \times 10^{-11}$ | $2.5 \times 10^{-09}$ | $2.2 \times 10^{-09}$ | LOC106603098, psma4, psme2, LOC106574401, sem1, LOC106608168, psmb7, LOC106560346, psma3, psmb6, psde, psmd8, LOC106570382, LOC106589310, adrm1, LOC106562664, psmd1, LOC100194646, prs10, LOC106569904, LOC106565847 |
| pyrimidine metabolism | 19 | $1.2 \times 10^{-04}$ | $1.5 \times 10^{-03}$ | $1.4 \times 10^{-03}$ | tyms, prim2, dut, cmpk2, rir2, rrm2, LOC106572454, pole3, LOC106606382, rir1, LOC106603918, ctps1, prim1, LOC106587512, pole, kith, polr1a, pola2, pole2 |
| oxidative phosphorylation | 18 | $1.6 \times 10^{-03}$ | $1.6 \times 10^{-02}$ | $1.4 \times 10^{-02}$ | LOC106585766, LOC106610322, LOC106605843, LOC106604229, LOC106604444, LOC106578938, atp5mc3, LOC106603475, LOC106607492, LOC106579942, atp5f1e, LOC106613744, LOC106600492, LOC106568019, LOC106569998, LOC106562286, LOC106573602, LOC106574141 |
| ribosome biogenesis in eukaryotes | 18 | $1.8 \times 10^{-06}$ | $3.1 \times 10^{-05}$ | $2.7 \times 10^{-05}$ | LOC106608971, nola2, gar1, LOC106604513, LOC106564653, LOC106588067, imp3, LOC106575044, fbrl, rd1, LOC106603897, if6, rpp30, nol5, LOC106583012, nol6, ran1, LOC106607124 |
| nucleotide excision repair | 13 | $3.9 \times 10^{-07}$ | $7.8 \times 10^{-06}$ | $6.9 \times 10^{-06}$ | LOC106580445, rfc5, LOC106585071, rpa3, pole3, rfc2, LOC106606382, LOC106612933, lig1, LOC106605109, rfc4, pole, pole2 |
| mismatch repair | 11 | $4.2 \times 10^{-09}$ | $1.0 \times 10^{-07}$ | $8.9 \times 10^{-08}$ | LOC106580445, rfc5, LOC106585071, rpa3, rfc2, LOC106606382, LOC106612933, lig1, LOC106605109, rfc4, msh2 |
| base excision repair | 8 | $3.2 \times 10^{-04}$ | $3.9 \times 10^{-03}$ | $3.4 \times 10^{-03}$ | LOC106580445, LOC106585071, pole3, LOC106606382, lig1, pole, fen1, pole2 |
| homologous recombination | 7 | $2.8 \times 10^{-03}$ | $2.6 \times 10^{-02}$ | $2.3 \times 10^{-02}$ | rad51, rpa3, sem1, LOC106606382, LOC106612933, LOC106605109, rad51c |

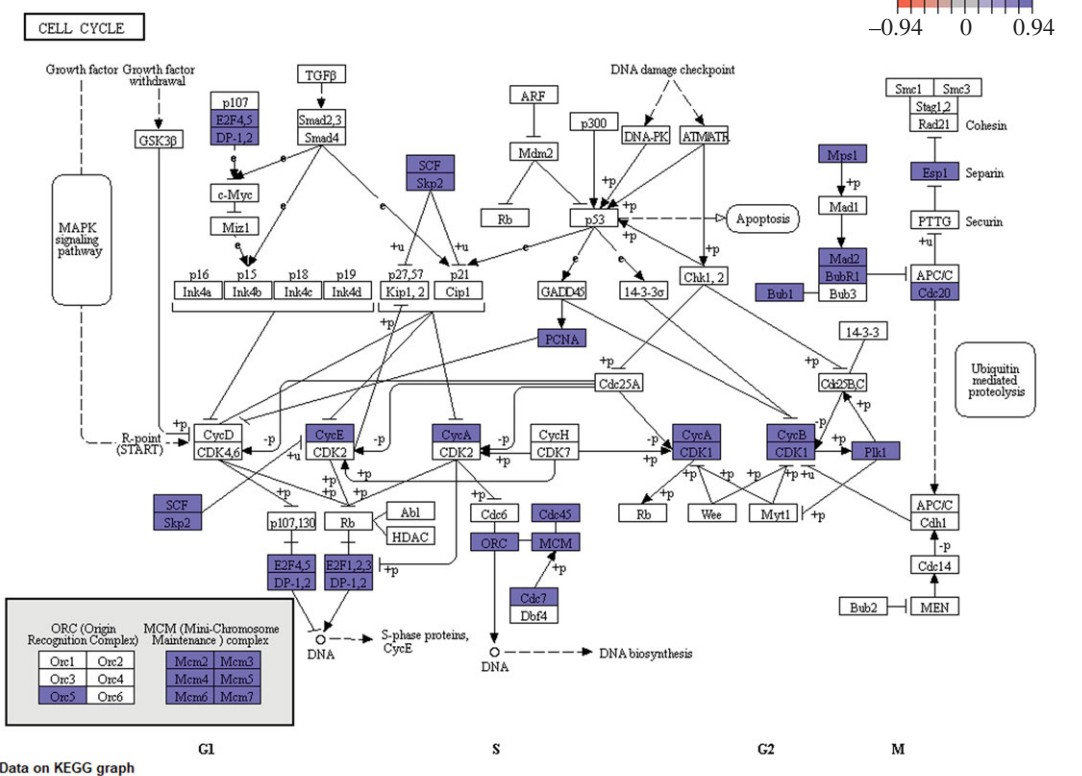

**Figure 4.** KEGG metabolic pathway map showing significantly differentially expressed genes within the cell cycle pathway for midgut control versus UCS. Red coloured genes are downregulated and purple coloured genes upregulated. All DE genes were upregulated in this pathway.

Under DNA duplication, the enteric epithelia requires strict control to avoid errors that may lead to the permanent alteration of the nucleotide sequence of the genome. The proliferating cell nuclear antigen (*pcna*) gene, which was upregulated in the UCS fish, is a well-known cell cycle marker that plays a central roles in cell cycle control and DNA replication [49], but is also involved in DNA excision repair, chromatin assembly and RNA transcription [50]. In the same context, the UCS upregulated the heterotrimeric *replication protein a* (*rpa*) gene, which is also involved in DNA replication, repair, recombination and telomere maintenance, coordinating the cellular response to DNA damage. Replication protein A, together with the upregulated recombination protein A (*rad51*), play an important role in the disposal of nuclear DNA waste.

The transcriptional data show that the fish require an increasing amount of energy to support midgut cell growth and renewal under stress, as indicated by enrichment of cell cycle and oxidative phosphorylation pathways. In fact, UCS fish showed an upregulated expression of 18 genes involved in the oxidative phosphorylation metabolic pathways (electronic supplementary material, figure S4), in which nutrients are oxidized to release energy required to produce adenosine triphosphate (ATP) inside mitochondria [51]. Interestingly, when nutrients are abundant, a similar upregulation of these pathways is observed in oncogene cells, which release energy to support cell growth and proliferation [52]. In addition to the regeneration of new tissue, the transcriptome data also indicated that removal of older/damaged tissue was occurring, as evidenced by the upregulation of 21 genes involved in the construction of both the proteolytic core 20S, and the regulatory subunit that constitute the proteasome protein complex (electronic supplementary material, figure S9).

Transcriptome results showed fewer differentially expressed genes and enriched pathways in the hindgut, compared to the midgut. Nevertheless, UCS caused a major downregulation of the differentially expressed genes in the hindgut tissue. Unfortunately, it was only possible to associate a small part of the differentially expressed set of genes to metabolic pathways (figure 3*b* and table 4) and, in addition, most of these genes were correlated at the cytokine–cytokine receptor interaction (figure 5). Cytokines are crucial intercellular regulators involved in immune and inflammatory responses [53]. Both cytokines and their receptors can be grouped by structure into different families.

**Table 4.** List of enriched (adjusted $p < 0.05$) KEGG pathways containing significantly differentially expressed genes (adjusted $p < 0.05$) in the Atlantic salmon hindgut.

| description | count | p-value | adjusted p | q-value | genes |
|---|---|---|---|---|---|
| cytokine–cytokine receptor interaction | 25 | $2.1 \times 10^{-08}$ | $2.3 \times 10^{-06}$ | $2.1 \times 10^{-06}$ | LOC106613637, LOC106608548, LOC106568165, LOC106606534, LOC106563690, LOC106573901, LOC106579503, ccr6, LOC106590519, LOC106606846, LOC106612744, LOC106591126, LOC106564390, LOC106611874, ccr9, il2rb, LOC106566619, LOC106588537, LOC106563758, LOC106601456, LOC106607522, LOC106607352, LOC106571057, LOC106564996, LOC106597311 |
| cellular senescence | 18 | $7.4 \times 10^{-05}$ | $4.0 \times 10^{-03}$ | $3.6 \times 10^{-03}$ | LOC106603043, LOC106607826, LOC106582436, pik3cd, LOC106582833, LOC106583670, LOC106588377, LOC106562702, LOC106565716, ccnb1, LOC106566708, LOC106573197, cdc2, LOC106608573, LOC106611545, LOC106587590, ccnb2, LOC106567114 |
| p53 signalling pathway | 10 | $1.4 \times 10^{-04}$ | $5.1 \times 10^{-03}$ | $4.7 \times 10^{-03}$ | 0C106607826, gtse1, ccnb1, LOC106566708, LOC106573197, cd82, LOC106583799, cdc2, ccnb2, rrm2 |

Of the downregulated genes in the cytokine–cytokine receptor interaction pathway (figure 5), the subfamily CC chemokines receptor *ccr6*, *ccr7* and *ccr9* and the ligand *ccl4* can be found, which are critical for the leucocyte migration during inflammation [54,55], and activation of the secondary immune response [56]. Interestingly, also in human clinical trials, using a *ccr9* antagonist to reduce *ccr9* gene expression has been modelled as an effective treatment in preventing inflammatory bowel disease, a chronic inflammation of the gastrointestinal tract that causes long-term tissue damages and often irreversible impairment of the structure and functions [57].

UCS was also associated with a downregulation of the *interleukin 6 signal transducer* (*il6st*) also known as *gp130* that, conversely to its name, is not only specific for Il6 but it is shared by all cytokines in the family [53]. In a similar manner, UCS downregulated the *interleukin 2 receptor subunit beta* (*il2rb*) and *gamma* (*il2rg*). IL2RG is an important signalling component of many interleukin receptors as a common signal transducer in their receptor complex, including those of interleukin -2, -4, -7–9, -15 and -21, and it is thus referred to as the common gamma chain. IL2RB, on the other hand, is only shared by IL2 and IL15. In mammals, stress can have biphasic actions on the immune system: when the stressor is acute it activates the acquired immune response and/or stimulates the production of immune mediators locally; on the contrary, chronic exposure to stress causes immune suppression [58]. Similarly, in this study, it appears that chronic stress led to a general downregulation of many cytokine pathways. Cortisol, above the other stress-regulated hormones, is recognized as one of the main factors inhibiting the release of proinflammatory cytokines (for an exhaustive review see Yada and Tort [59]). We hypothesize that under a chronic stress regime, the expression pattern just

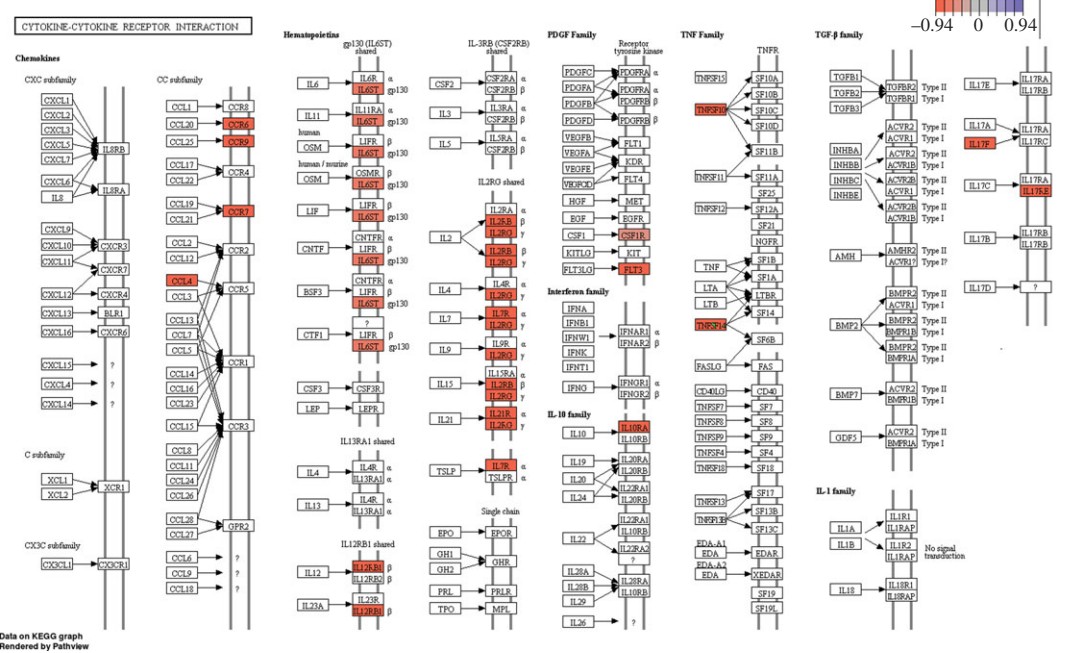

**Figure 5.** KEGG metabolic pathway map showing significantly differentially expressed genes within the cytokine–cytokine receptor interaction pathway for hindgut control versus UCS. Red coloured genes are downregulated and purple coloured genes upregulated. All DE genes were downregulated in this pathway.

described would prevent energy expenditure on one side, but on the other reduce the fish immune response and inflammations. Therefore, this coping mechanism could potentially cause a reduced immune resistance of fish and a major susceptibility to pathogen infection.

More intricate is the link between UCS and the identified differentially expressed genes involved in the p53 signalling and cellular senescence (electronic supplementary material, figures S3, S8). In mammals, the activation of the p53 pathways are induced by several stress signals, including DNA damage, oxidative stress and activated apoptosis. The p53 protein is a transcriptional activator of several p53-regulated genes. Interestingly, though many genes of the p53 pathway were differentially expressed in the hindgut, the *p53* gene itself was not. Unfortunately, it is not possible to say which kind of stress signal activated genes that are associated with this pathway.

# 5. Conclusion

Midgut and hindgut transcription and histological analyses were conducted on Atlantic salmon following three weeks of UCS. Histological analysis did not show any significant effect on morphometric parameters. In the midgut, 1030 genes were differentially expressed following UCS, of which 329 were downregulated and 701 upregulated. Of these 279 genes were involved in 13 pathways. Upregulated pathways include tissue repair pathways such as cell cycle, DNA replication, mismatch repair, nucleotide excision repair, base excision repair, ribosome and ribosome biogenesis, RNA transport, spliceosome, pyrimidine metabolism, proteasome, homologous recombination and oxidative phosphorylation. In the hindgut, UCS resulted in a regulation of 591 differentially expressed genes where 426 were downregulated and 165 upregulated. A total of 53 genes were related to three pathways. Downregulated genes include cellular senescence pathways, p53 signalling and cytokine–cytokine receptor pathways.

Although the data represents a snapshot of the gut condition at the end of a 23-day trial, results indicate that salmon were at least partly habituating to the UCS treatment. Even though no morphological differences were seen, the transcriptional patterns show that the fish were still affected by the UCS. In the midgut, the main upregulation was related to cell growth and repair, while in the hindgut there were indications of the activated apoptotic pathway, reduced cell repair and inhibited immune/inflammatory capacity. This may be the trade-off between habituating to UCS and health resilience.

Ethics. This work was conducted in accordance with the laws and regulations controlling experiments and procedures on live animals in Norway and was approved by the Norwegian Animal Experiment Committee (Forsøksdyrutvalget, 11.12.2012).

Data accessibility. Fastq files were uploaded to the NCBI sequence read archive (SRA). SRA accession number is SRP169832. R code for the analysis pipeline can be found in the supplementary material.

Authors' contributions. S.D.L. analysed samples and wrote the paper; A.M. designed and conducted the experiment and collaborated to write the manuscript. M.-A.O. analysed samples. P.W. contributed with the transcriptome data analyses and writing the paper. T.B. performed histology analyses. S.R.S. collaborate in the data analysis; R.E.O. designed and conducted the experiment, processed tissue samples and collaborated to write the manuscript.

Competing interests. We declare we have no competing interest.

Funding. This project was funded by the Norwegian University of Science and Technology. The trial was funded by the European Community's Seventh Framework Programme [FP7/2010–2014] under grant agreement no. 265957—COPEWELL.

Acknowledgements. We would like to thank the staff at the IMR Research station in Matre for their help in experimental design and sampling, in particular Ivar Helge Matre, Kåre Storsæter, Britt Sværen, Karen Anita Kvestad and Tone Vågseth.

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
