## [Reviewer comments · Royal Society Open Science]

Review History

RSOS-191480.R0 (Original submission)

Review form: Reviewer 1

Is the manuscript scientifically sound in its present form?

Yes

Are the interpretations and conclusions justified by the results?

Yes

Is the language acceptable?

Yes

Do you have any ethical concerns with this paper?

No

Have you any concerns about statistical analyses in this paper?

No

Recommendation?

Accept with minor revision (please list in comments)

Comments to the Author(s)

This is a nicely presented manuscript with some interesting findings, relevant to better understand stress responses in Atlantic salmon. I would suggest some minor revision to clarify some statements made throughout the text, as outlined below.

Overall comments

1. I found that for several statements made in the introduction and discussion, no reference is mentioned. I would recommend reviewing these sections again and adding references in the appropriate places (examples: ll 38-39, ll 375-356).
2. Although no specific stress factor was tested, but rather random, changing events, it would be helpful if the rationale of these stressors was explained in context of the physiology of the fish, e.g. "natural" and "unnatural" stressors and how the fish's system may be inherently well equipped to deal with this. This could be put into the discussion.
3. The results from the histology suggest no difference between stressed and unstressed fish for both gut sections for the features analysed. I would suggest looking at more parameters that cover the whole gut section as sometimes damage has been found to be localised in certain areas. Furthermore, features not analysed here such as supranuclear vacuole appearance have been found to be good markers for tissue damage. I would recommend considering the scoring system proposed by Uran et al (2009) as additional parameters.
4. The manuscript would benefit from careful proofreading - in several places punctuations are missing. Furthermore, there are inconsistencies in the formatting of "padj" and referencing authors in the text.

Methods section

1. It is not clear if you have assigned all identified genes a gene symbol (HGNC) prior to gene ontology and pathway analysis. Only the minority of loci of the salmon genome is annotated with a gene symbol, but the majority is only annotated with a LOC number, therefore you would lose a lot of differentially expressed genes in this analysis if you do not assign them a symbol.
2. You mention numbers for up and downregulated genes for each gut section, but it would be good if you would also mention how many of these are unique or duplicated genes and how many of these genes were fed into gene ontology and pathway analysis and if you analysed up and downregulated genes separately or together as differentially expressed genes.
3. Furthermore it would be useful to know what cut-off you used, you mention a $\text{padj} < 0.05$ but did you also filter by fold-change?

Results

1. As mentioned above, when you provide the numbers for up and downregulated genes, it is useful to know what cut-offs you used.
2. Is the cortisol graph meant to go into the main text or supplementary material? Should be mentioned in the appropriate paragraph.
3. Table 3 and 4: it would be useful to translate the LOC numbers into gene names or symbols so the reader can easily understand what genes are associated with the kegg term. Also some indication if these are up or downregulated would be useful.

Discussion

1. The cortisol graph shows a dip in cortisol in the stressed group on day 5 relative to the other time points. It would be interesting if you could discuss why this may be in a few sentences.
2. Do you have any suggestions why tissue repair was evident from transcriptomic data and not histological data? May that be due to the only experimental timepoint being when there was already some signs to acclimatisation?
3. Could you please expand on the agenda of feeding the fish so shortly after having experienced stress? Would the fish have consumed food like the control fish if the feeding had taken place, let's say 2 hours after stress, therefore not had altered growth parameters? I understand it is

impossible to include everything in an experiment, but I think it makes an interesting point for the discussion.

4. Lastly, I would suggest to check that you included all up-to-date references on this topic, a lot of references seem fairly old, and even when relevant, should be supplemented by the inclusion of the newest findings.

Review form: Reviewer 2

Is the manuscript scientifically sound in its present form?

No

Are the interpretations and conclusions justified by the results?

No

Is the language acceptable?

Yes

Do you have any ethical concerns with this paper?

No

Have you any concerns about statistical analyses in this paper?

Yes

Recommendation?

Major revision is needed (please make suggestions in comments)

Comments to the Author(s)

Manuscript ID: RSOS-191480

This manuscript reports on a study conducted in 2013 that examined the mid and hindgut transcriptome profile of Atlantic salmon parr subjected to a 3-week unpredictable chronic stress. The study reports on results from samples collected at the last day of the study and compares control and stressed group. Ideally this study should have included in the transcriptome analysis more fish and sampling times. A comparison with samples collected at the start of the experiment would be beneficial to understand overtime changes.

Major comments,

Please provide clearly the objectives of the study. The introduction is focused in explaining stress, chronic stress, and does not really cover transcriptome studies in fish during stress, and is somehow disconnected from the discussion.

The sampling protocol needs more clarification. A small sample size was used in this study (3 fish from the control group and 4 fish from the stressed group), and it is not clear where the samples are coming from. Are they coming from different tanks? Mid and hindgut samples were collected from the same fish. For example, control MID 1 and control HIND 1 are samples from the same fish. Ideally this study should have included in the transcriptomic analysis more fish and sampling times. A comparison with samples collected at the start of the experiment would be beneficial to understand overtime changes.

Provide the following information (clarify in the material and methods): were the gene counts filtered to only keep genes that were present in all replicate samples of the same treatment? Did you keep only the genes that had a normalized expression over for example 1 count per million? Normalization of genes by counts per million controls for differences in library size, reducing bias.

Summarize the gene expression results. The authors should include a table demonstrating the top genes up and down regulated, and possibly present the genes that were similarly up and down regulated in both mid and hind gut.

The manuscript is well written, but some elements of the discussion can be moved to intro or removed, as it is rather lengthy. The discussion would also benefit of exploring further the lack of significant results on histo when compared to gene expression. Is it possible that longer periods of stress would have caused changes on the morphometric parameters? Or is it possible that you did not collect samples at the right time to show morphological changes? (discussion L321-338). What is the significance of all these transcriptomic/pathways changes if no morphological changes occurred? Are these transcriptional changes consistent in all samples?

Decision letter (RSOS-191480.R0)

04-Dec-2019

Dear Ms Løvmo,

The editors assigned to your paper ("Mid and hindgut transcriptome profiling analysis of Atlantic salmon (*Salmon salar*) under unpredictable chronic stress") have now received comments from reviewers. We would like you to revise your paper in accordance with the referee and Associate Editor suggestions which can be found below (not including confidential reports to the Editor). Please note this decision does not guarantee eventual acceptance.

Please submit a copy of your revised paper before 27-Dec-2019. Please note that the revision deadline will expire at 00.00am on this date. If we do not hear from you within this time then it will be assumed that the paper has been withdrawn. In exceptional circumstances, extensions may be possible if agreed with the Editorial Office in advance. We do not allow multiple rounds of revision so we urge you to make every effort to fully address all of the comments at this stage. If deemed necessary by the Editors, your manuscript will be sent back to one or more of the original reviewers for assessment. If the original reviewers are not available, we may invite new reviewers.

- Data accessibility

If you wish to submit your supporting data or code to Dryad (<http://datadryad.org/>), or modify your current submission to dryad, please use the following link:
<http://datadryad.org/submit?journalID=RSOS&manu=RSOS-191480>

- Competing interests

- Authors' contributions

- Acknowledgements

- Funding statement

on behalf of Dr Michael Tobler (Associate Editor) and Professor Kevin Padian (Subject Editor)
openscience@royalsociety.org

Associate Editor's comments (Dr Michael Tobler):

We have received the feedback from two reviewers, both of which agreed about the merits of this study. Although there were some concerns about low sample size, I think this manuscripts will fit within the scope of RSOS upon revision. Both reviewers provided constructive feedback that should help the authors to improve their manuscript. I agree with the reviewers that the manuscript would benefit from a broader conceptual context that is rooted on previously published studies. In addition, clarifying some gaps in the methods and analytical approaches is necessary.

Reviewers' Comments to Author:

Reviewer: 1

Comments to the Author(s)

This is a nicely presented manuscript with some interesting findings, relevant to better understand stress responses in Atlantic salmon. I would suggest some minor revision to clarify some statements made throughout the text, as outlined below.

Overall comments

1. I found that for several statements made in the introduction and discussion, no reference is mentioned. I would recommend reviewing these sections again and adding references in the appropriate places (examples: ll 38-39, ll 375-356).
2. Although no specific stress factor was tested, but rather random, changing events, it would be helpful if the rationale of these stressors was explained in context of the physiology of the fish, e.g. "natural" and "unnatural" stressors and how the fish's system may be inherently well equipped to deal with this. This could be put into the discussion.
3. The results from the histology suggest no difference between stressed and unstressed fish for both gut sections for the features analysed. I would suggest looking at more parameters that cover the whole gut section as sometimes damage has been found to be localised in certain areas. Furthermore, features not analysed here such as supranuclear vacuole appearance have been found to be good markers for tissue damage. I would recommend considering the scoring system proposed by Uran et al (2009) as additional parameters.
4. The manuscript would benefit from careful proofreading - in several places punctuations are missing. Furthermore, there are inconsistencies in the formatting of "padj" and referencing authors in the text.

Methods section

1. It is not clear if you have assigned all identified genes a gene symbol (HGNC) prior to gene ontology and pathway analysis. Only the minority of loci of the salmon genome is annotated with a gene symbol, but the majority is only annotated with a LOC number, therefore you would lose a lot of differentially expressed genes in this analysis if you do not assign them a symbol.
2. You mention numbers for up and downregulated genes for each gut section, but it would be good if you would also mention how many of these are unique or duplicated genes and how many of these genes were fed into gene ontology and pathway analysis and if you analysed up and downregulated genes separately or together as differentially expressed genes.
3. Furthermore it would be useful to know what cut-off you used, you mention a $\text{padj} < 0.05$ but did you also filter by fold-change?

Results

1. As mentioned above, when you provide the numbers for up and downregulated genes, it is useful to know what cut-offs you used.
2. Is the cortisol graph meant to go into the main text or supplementary material? Should be mentioned in the appropriate paragraph.
3. Table 3 and 4: it would be useful to translate the LOC numbers into gene names or symbols so the reader can easily understand what genes are associated with the kegg term. Also some indication if these are up or downregulated would be useful.

Discussion

1. The cortisol graph shows a dip in cortisol in the stressed group on day 5 relative to the other time points. It would be interesting if you could discuss why this may be in a few sentences.
2. Do you have any suggestions why tissue repair was evident from transcriptomic data and not histological data? May that be due to the only experimental timepoint being when there was already some signs to acclimatisation?
3. Could you please expand on the agenda of feeding the fish so shortly after having experienced stress? Would the fish have consumed food like the control fish if the feeding had taken place, let's say 2 hours after stress, therefore not had altered growth parameters? I understand it is impossible to include everything in an experiment, but I think it makes an interesting point for the discussion.
4. Lastly, I would suggest to check that you included all up-to-date references on this topic, a lot of references seem fairly old, and even when relevant, should be supplemented by the inclusion of the newest findings.

Reviewer: 2

Comments to the Author(s)

Manuscript ID: RSOS-191480

This manuscript reports on a study conducted in 2013 that examined the mid and hindgut transcriptome profile of Atlantic salmon parr subjected to a 3-week unpredictable chronic stress. The study reports on results from samples collected at the last day of the study and compares control and stressed group. Ideally this study should have included in the transcriptome analysis more fish and sampling times. A comparison with samples collected at the start of the experiment would be beneficial to understand overtime changes.

Major comments,

Please provide clearly the objectives of the study. The introduction is focused in explaining stress, chronic stress, and does not really cover transcriptome studies in fish during stress, and is somehow disconnected from the discussion.

The sampling protocol needs more clarification. A small sample size was used in this study (3 fish from the control group and 4 fish from the stressed group), and it is not clear where the samples are coming from. Are they coming from different tanks? Mid and hindgut samples were collected from the same fish. For example, control MID 1 and control HIND 1 are samples from the same fish. Ideally this study should have included in the transcriptomic analysis more fish and

sampling times. A comparison with samples collected at the start of the experiment would be beneficial to understand overtime changes.

Provide the following information (clarify in the material and methods): were the gene counts filtered to only keep genes that were present in all replicate samples of the same treatment? Did you keep only the genes that had a normalized expression over for example 1 count per million? Normalization of genes by counts per million controls for differences in library size, reducing bias.

Summarize the gene expression results. The authors should include a table demonstrating the top genes up and down regulated, and possibly present the genes that were similarly up and down regulated in both mid and hind gut.

The manuscript is well written, but some elements of the discussion can be moved to intro or removed, as it is rather lengthy. The discussion would also benefit of exploring further the lack of significant results on histo when compared to gene expression. Is it possible that longer periods of stress would have caused changes on the morphometric parameters? Or is it possible that you did not collect samples at the right time to show morphological changes? (discussion L321-338). What is the significance of all these transcriptomic/pathways changes if no morphological changes occurred? Are these transcriptional changes consistent in all samples?

Author's Response to Decision Letter for (RSOS-191480.R0)

See Appendix A.

RSOS-191480.R1 (Revision)

Review form: Reviewer 2

Is the manuscript scientifically sound in its present form?

Yes

Are the interpretations and conclusions justified by the results?

Yes

Is the language acceptable?

Yes

Do you have any ethical concerns with this paper?

No

Have you any concerns about statistical analyses in this paper?

No

Recommendation?

Accept with minor revision (please list in comments)

Comments to the Author(s)

The manuscript should be once more very carefully proofread for proper wording, grammar and punctuation.

Examples:

Line 147. Missing a period at the end of the sentence. Also should the number of that reference be between brackets?

Line 152: Sections we stained with.... Should be Sections were stained....

Line 179: change respectfully, with respectively.

Decision letter (RSOS-191480.R1)

24-Jan-2020

Dear Ms Løvmo:

On behalf of the Editors, I am pleased to inform you that your Manuscript RSOS-191480.R1 entitled "Mid and hindgut transcriptome profiling analysis of Atlantic salmon (*Salmo salar*) under unpredictable chronic stress" has been accepted for publication in Royal Society Open Science subject to minor revision in accordance with the referee suggestions. Please find the referees' comments at the end of this email.

The reviewers and Subject Editor have recommended publication, but also suggest some minor revisions to your manuscript. Therefore, I invite you to respond to the comments and revise your manuscript.

- Ethics statement

- Data accessibility

<http://datadryad.org/submit?journalID=RSOS&manu=RSOS-191480.R1>

- Competing interests

- Authors' contributions

- Acknowledgements

- Funding statement

Because the schedule for publication is very tight, it is a condition of publication that you submit the revised version of your manuscript before 02-Feb-2020. Please note that the revision deadline will expire at 00.00am on this date. If you do not think you will be able to meet this date please let me know immediately.

Kind regards,
Lianne Parkhouse
Editorial Coordinator
Royal Society Open Science
openscience@royalsociety.org

on behalf of Dr Michael Tobler (Associate Editor) and Kevin Padian (Subject Editor)
openscience@royalsociety.org

Associate Editor Comments to Author (Dr Michael Tobler):

The authors have satisfactorily addressed previous reviewer concerns, and this manuscript is acceptable for publication after some minor revisions. I have a few additional comments that might help to improve the clarity of the figures:

Figure 2: The colors and symbols are redundant. I would suggest to use the same color for $p < 0.05$ and $p < 0.01$ irrespectively if genes are up or down regulated. Also, please avoid red and green in the same graph to make sure readers with color blindness can interpret the results.

Figure 4 & 5: If possible, avoid the red-green gradient (see comment above).

Reviewer comments to Author:

Reviewer: 2
Comments to the Author(s)

The manuscript should be once more very carefully proofread for proper wording, grammar and punctuation.

Examples:

Line 147. Missing a period at the end of the sentence. Also should the number of that reference be between brackets?

Line 152: Sections we stained with.... Should be Sections were stained....

Line 179: change respectfully, with respectively.

Author's Response to Decision Letter for (RSOS-191480.R1)

See Appendix B.

Decision letter (RSOS-191480.R2)

31-Jan-2020

Dear Ms Løvmo,

It is a pleasure to accept your manuscript entitled "Mid and hindgut transcriptome profiling analysis of Atlantic salmon (*Salmo salar*) under unpredictable chronic stress" in its current form for publication in Royal Society Open Science.

Kind regards,
Lianne Parkhouse
Editorial Coordinator
Royal Society Open Science
openscience@royalsociety.org

on behalf of Dr Michael Tobler (Associate Editor) and Kevin Padian (Subject Editor)
openscience@royalsociety.org

Appendix A

We thank the reviewer for taking the time to review our manuscript and giving much appreciated comments that have improved the manuscript greatly. Our reply to the reviewer is written in italics under the comments.

Reviewer: 1

Comments to the Author(s)

This is a nicely presented manuscript with some interesting findings, relevant to better understand stress responses in Atlantic salmon. I would suggest some minor revision to clarify some statements made throughout the text, as outlined below.

Overall comments

1. I found that for several statements made in the introduction and discussion, no reference is mentioned. I would recommend reviewing these sections again and adding references in the appropriate places (examples: ll 38-39, ll 375-356).

R: We thank the reviewer for pointing out our mistakes and have review the introduction and discussion and updated the references

- *Rearranged references in line 39-40*
- *Added reference to line 48, Shreck and Tort 2016)*
- *Added reference line 73, Wilner, muscat and Papp 1992)*
- *Added reference line 77, Piato, Capiotti et al. 2011)*
- *Added references line 383, Niklasson et al. 2011; Segner et al. 2012; Sundh et al. 2010*
- *Added reference line 388, Niklasson et al. 2011; Sundh et al. 2010*

2. Although no specific stress factor was tested, but rather random, changing events, it would be helpful if the rationale of these stressors was explained in context of the physiology of the fish, e.g. "natural" and "unnatural" stressors and how the fish's system may be inherently well equipped to deal with this. This could be put into the discussion.

R: The rationale behind the stressors in the UCS regime is mostly to induce several different stressful events that prevented fish from adapting over time, however without exposing the fish to severe physical treatment (introduction, line 69-75). The stressor in this trail are not intended to mimic "natural" or "unnatural" events, but only to develop a model of UCS in order to asses a prolonged stress response.

3. The results from the histology suggest no difference between stressed and unstressed fish for both gut sections for the features analysed. I would suggest looking at more parameters that cover the whole gut section as sometimes damage has been found to be localised in certain areas. Furthermore, features not analysed here such as supranuclear vacuole appearance have been found to be good markers for tissue damage. I would recommend considering the scoring system proposed by Uran et al (2009) as additional parameters.

R: As suggested by the reviewer, we have gone back and reviewed all histological pictures of the intestinal segments for vacuolization, and we can see no difference between the UCS group and the

control group. We also don't see any pattern in local damage between the groups in the sections that are made. The results are not included in the manuscript.

4. The manuscript would benefit from careful proofreading - in several places punctuations are missing. Furthermore, there are inconsistencies in the formatting of "padj" and referencing authors in the text.

R: We thank the reviewer for pointing this out, and have proofread the manuscript again. We have also changed all p.adj/p.adjust to adjusted p

5. It is not clear if you have assigned all identified genes a gene symbol (HGNC) prior to gene ontology and pathway analysis. Only the minority of loci of the salmon genome is annotated with a gene symbol, but the majority is only annotated with a LOC number, therefore you would lose a lot of differentially expressed genes in this analysis if you do not assign them a symbol.

R: Great point. We did use Entrez gene IDs for GO and KEGG analysis and as Entrez are foundational identifiers in the KEGG and GO databases, and Atlantic salmon is a KEGG organism, all genes were used in GO/KEGG analysis. We annotated Entrez IDs to gene symbols and description for the manuscript, as Entrez IDs are purely numerical and we felt gene symbols present better in the manuscript (even if most are 'LOC..'). In supplementary table 3 we provide all 3 identifiers (Entrez ID, gene symbol and gene description).

The following has now been added the above information to the 'Differential expression analysis and functional annotations' subsection in the 'Materials and methods' section

line 213-217: "Genes were initially annotated with Entrez gene identifiers and were subsequently annotated to gene symbols and gene descriptions. While gene symbols are presented in this paper, Entrez IDs were used for KEGG and GO enrichment analysis. Supplementary tables of DE genes and enriched pathways contain all 3 identifiers for each associated gene – Entrez ID, gene symbol and gene description."

6. You mention numbers for up and downregulated genes for each gut section, but it would be good if you would also mention how many of these are unique or duplicated genes and how many of these genes were fed into gene ontology and pathway analysis and if you analysed up and downregulated genes separately or together as differentially expressed genes.

R: There were around 29% duplicated genes in the total list of genes (supplementary tables 4 and 3) per gut region. Almost all of these were tRNA and none of the DE genes were duplicates. We have amended the manuscript (Results section) as follows

line 292-294: "Annotation to the 44868 Atlantic salmon reference genes produced around 29% of duplicated Entrez gene identifiers, but the vast majority (>99%) of these duplicates were transfer RNA. None of the DE genes were duplicates."

We did GO and KEGG analysis using all the DE genes (i.e. up and downregulated combined). The reason for this is twofold: 1) analysis was based on an over-representation test, for which statistical power is reduced as the number of genes input is reduced and 2) we included which genes were up and downregulated in the KEGG pathway maps (figure 4 and 5) and by using all genes we can indeed see that some pathways are entirely up or downregulated. We agree that we did not clearly mention this in the manuscript though, so have made the following amendments.

In the manuscript (Materials and methods) we changed the following from (line 231-232):

*“we identified enriched pathways by performing an over-representation test **on our DE gene list**”*

To:

*“we identified enriched pathways by performing an over-representation test **based on the total set of DE genes per treatment**”*

7. Furthermore it would be useful to know what cut-off you used, you mention a $p_{adj} < 0.05$ but did you also filter by fold-change?

Indeed, to use or not use a fold change cutoff has been the source of much discussion and examination within our group. We recognise that the likelihood of false positive DE genes is increased as fold change difference decreases, but we also noted that fold change cutoff varies considerably between publications and specific fold change cutoffs are almost always selected without semi-arbitrarily and without statistical justification (1, 1.2, 1.5, 2?). It is for this reason that we finally decided on using just the statistical significance test itself (i.e. false discovery adjusted p values) and no fold change cutoff.

We added the following to the ‘Differential expression analysis and functional annotations’ subsection in the ‘Materials and methods’ section (line 225-226):

“We did not include a fold change cutoff but relied on DESeq2 to identify which genes were significantly DE, as DESeq2 controls for false positives and is sensitive to small, true differences.”

8. As mentioned above, when you provide the numbers for up and downregulated genes, it is useful to know what cut-offs you used

R: Please see point 7, above.

9. Is the cortisol graph meant to go into the main text or supplementary material? Should be mentioned in the appropriate paragraph.

R: Thank you for pointing this out, the cortisol graph is in the supplementary material as figure 15. This is now added to the manuscript (line 240).

10. Table 3 and 4: it would be useful to translate the LOC numbers into gene names or symbols so the reader can easily understand what genes are associated with the kegg term. Also some indication if these are up or downregulated would be useful.

R: Unfortunately, those LOC numbers are the official gene symbols for Atlantic salmon. The only explanatory information that can be associated with these gene symbols is the full gene descriptions, which are too long to be added to table 3 and 4. However, the associated gene description, Entrez ID as well as the log fold change and adjusted p values are in the supplementary tables 1 and 2. We have added the following to the manuscript

Results section, line 327 -332: "Note that tables 5 and 6 show the Atlantic salmon gene symbol for each DE gene per enriched pathway. As the Atlantic salmon reference genome has been constructed relatively recently, most of these are 'placeholder' symbols in the form of 'LOC....' and are associated with homologous genes in other species. More information for each of these gene symbols can be found in supplementary tables 1 and 2, including Entrez ID, full gene description, log2 fold change and false-discovery adjusted p values."

Discussion

11. The cortisol graph shows a dip in cortisol in the stressed group on day 5 relative to the other time points. It would be interesting if you could discuss why this may be in a few sentences.

R: This dip in cortisol have been discussed in the previous published article Madaro et al. 2015: "However, on day 5 of the experiment, the UCS group displayed a remarkable low cortisol production post-stress. Perhaps the UCS together with the high frequency of samplings during the first days of the experiment resulted in an exhaustion of the HPI axis and thus a failure to mount a proper cortisol response following stress."

12. Do you have any suggestions why tissue repair was evident from transcriptomic data and not histological data? May that be due to the only experimental timepoint being when there was already some signs to acclimatisation?

R: Yes. As discussed in the manuscript, we suggest that the lack of histological effects from the UCS comes from the Atlantic salmon's ability to adapt to stressors. The upregulation of tissue repair in the midgut suggests that the intestine is still affected by the UCS, but based on the histological results the fish is able to maintain normal intestinal function. In hindsight, we see that additional samplings for histology at earlier time-points would probably have given a broader understanding of the fish's reaction to UCS.

We have amended the manuscript to make this more clear: Discussion line 381-384 and 393-396, and conclusion line 494-500

13. Could you please expand on the agenda of feeding the fish so shortly after having experienced stress? Would the fish have consumed food like the control fish if the feeding had taken place, let's say 2 hours after stress, therefore not had altered growth parameters? I understand it is impossible to include everything in an experiment, but I think it makes an interesting point for the discussion.

R: The reviewer's comment is interesting. If the design of the experiment was different, and there was only one or maybe two stressors per day, it is possible that delaying feeding 2h after each stress episode would give the fish enough time to recover and limit the effect of stress on appetite and growth. But the fish would also most likely habituate to the stress in a shorter time. In this trial, stress episodes were set for three times a day in a random manner in order to reduce habituation to the stressors, and we believe that giving them little more time between stress and feeding would not affect much of the growth, at least in the time period of the study.

However, there is also a logistical problem with delaying the feeding time, and that is also the main reason for having such short time between stress and feeding. Delaying feeding would either:

- *Reduce time available for the fish to eat*
- *Reduce time between feeding and stress; meaning the fish would be in a postprandial state under stress which could cause vomiting and discomfort for the fish.*
- *Compromise feed collection; stress and feed collection would overlap*
- *Delay the last feeding into the dark period of the day, as the fish were on a 12h light:12h dark light cycle.*

14. Lastly, I would suggest to check that you included all up-to-date references on this topic, a lot of references seem fairly old, and even when relevant, should be supplemented by the inclusion of the newest findings.

R: We have gone through the reference list and updated it.

Reviewer: 2

Comments to the Author(s)

We thank the reviewer for taking the time to review our manuscript and giving much appreciated comments that have improved the manuscript greatly. Our reply to the reviewer is written in italics under the comments.

Manuscript ID: RSOS-191480

This manuscript reports on a study conducted in 2013 that examined the mid and hindgut transcriptome profile of Atlantic salmon parr subjected to a 3-week unpredictable chronic stress. The study reports on results from samples collected at the last day of the study and compares control and stressed group. Ideally this study should have included in the transcriptome analysis more fish and sampling times. A comparison with samples collected at the start of the experiment would be beneficial to understand overtime changes.

Major comments,

1. Please provide clearly the objectives of the study. The introduction is focused in explaining stress, chronic stress, and does not really cover transcriptome studies in fish during stress, and is somehow disconnected from the discussion.

R: We thank the reviewer for taking the time to review our manuscript and for the suggestions that will improve the manuscript. We have now added our aim to the introduction.

line 83: "The aim of this study was to investigate the effect of unpredictable chronic stress on Atlantic salmon smolt intestine"

We have added some references to the limited number of transcriptional studies on stress and the intestine.

*Line 59-65: "Furthermore, information on the transcriptional mechanisms activated during stress in fish intestine are very few. For example, some studies have also investigated the transcriptional profile of the fish intestine after stress (19). In Asian Seabass (*Lates calcarifer*), partial repression of the intestinal immune system is seen after being challenged by pathogen or immune modulators (20). Sending Japanese Medaka to the international space station showed a higher impact on the intestine compared to other organs in the RNA expression profile, while almost no difference were seen in morphology (21)."*

2. The sampling protocol needs more clarification. A small sample size was used in this study (3 fish from the control group and 4 fish from the stressed group), and it is not clear where the samples are coming from. Are they coming from different tanks? Mid and hindgut samples were collected from the same fish. For example, control MID 1 and control HIND 1 are samples from the same fish. Ideally this study should have included in the transcriptomic analysis more fish and sampling times. A comparison with samples collected at the start of the experiment would be beneficial to understand overtime changes.

R: We see that the text needed some clarification regarding sampling number under Methods, and number of individuals samples have been removed from "experimental design" (line 110-112 in first draft of the manuscript), and are now clearly stated under "Histological analysis" and "RNA extraction"

line: 149-150: "Midgut and hindgut samples were taken from five and 6 fish per treatment (N=5/6, from all three replicates)"

line 178-180: "Three fish from the control group and four fish from the UCS group selected from two and three replicates, respectfully, and samples from midgut and hindgut were used to construct the sequencing libraries (14 samples in total)."

We also agree with the reviewer, and see that additional samplings at more time-points would have been more beneficial in understanding how the fish cope with the UCS regime. However, we attempted to find the right balance between available resources and scientific output.

3. Provide the following information (clarify in the material and methods): were the gene counts filtered to only keep genes that were present in all replicate samples of the same treatment? Did you keep only the genes that had a normalized expression over for example 1 count per million? Normalization of genes by counts per million controls for differences in library size, reducing bias.

R: DESeq2 requires a count table of raw read counts per feature (e.g. gene) as input. This is because it performs an internal normalization, correcting for library size and RNA composition bias, based on raw counts. It also removes low count genes. We agree that we had not explained this in sufficient detail and have updated the Materials and methods section to clarify this:

Line 220-223 "DESeq2 requires a count table of raw read counts per gene as input and performs internal normalization for both sequencing depth (library size) and RNA composition based on the geometric mean per gene across all treatment samples."

4. Summarize the gene expression results. The authors should include a table demonstrating the top genes up and down regulated, and possibly present the genes that were similarly up and down regulated in both mid and hind gut.

R: We had included a table of genes that are up and downregulated in both midgut and hindgut in the supplementary section, but we neglected to add a reference to this in the manuscript. Thank you for spotting this. We've amended the manuscript (Results section). Regarding including top up and down regulated genes in the article, we think it will be confusing to the reader as we do not discuss single up and downregulated genes but focus on the enriched pathways in the manuscript, and therefore choose to only list them in the supplementary data.

Line 291-302: "Gene identification was based on the Atlantic salmon ICSASG_v2 reference genome. Annotation to the 44868 Atlantic salmon reference genes produced around 29% of duplicated Entrez gene identifiers, but the vast majority (>99%) of these duplicates were transfer RNA. None of the DE genes were duplicates. Of the 1030 genes that were differentially expressed (DE) in the midgut, 329 were downregulated and 701 upregulated (Fig 2A, supplementary table 4). In the hindgut 591 genes were differentially expressed, of which 426 were downregulated and 165 upregulated (Fig 2B, supplementary table 3). There were 114 genes that were concordantly differentially expressed in both midgut and hindgut (Supplementary table 6). The top 2 concordant genes, both strongly downregulated, were rho-associated protein kinase 2-like (-6.10 log₂fc in hindgut and -5.32 log₂fc in midgut) and von Willebrand factor A domain-containing protein 7-like (-4.43 and -5.03 log₂fc in hindgut and midgut respectively).

5. The manuscript is well written, but some elements of the discussion can be moved to intro or removed, as it is rather lengthy. The discussion would also benefit of exploring further the lack of significant results on histo when compared to gene expression.

R: We agree with the reviewer, and have moved part of the discussion on effect of stress now into the introduction (line 49-59) and removed some parts (first draft of the manuscript, line 365-368, 383-385, 400-404)

The lack of histological effects is more clearly elaborated now in the discussion , line 381-384 and 393-396, and conclusion line 494-500

6. Is it possible that longer periods of stress would have caused changes on the morphometric parameters? Or is it possible that you did not collect samples at the right time to show morphological changes? (discussion L321-338).

R: The reviewer questions are legit. If we take as true that fish were recovering appetite at the end of the trial due to the habituation to stressors, maybe a longer stress would not show changes in the morphometric parameters. As for the timing of sampling, it is possible that several sampling at the beginning of the trial, may have displayed more defined morphological differences between the groups.

7. What is the significance of all these transcriptomic/pathways changes if no morphological changes occurred? Are these transcriptional changes consistent in all samples?

R: We believe that even though the fish started to habituating to the UCS regime (as indicated by appetite and histology), the transcriptional profile of the intestine in this study shows indications that mechanisms for coping stress were still ongoing in the attempt to assure gut health.

Updated conclusion, line 494-500

Appendix B

31-Jan-2020

Dear Lianne Parkhaouse, Dr Michael Tobler and Kevin Pardian

We would like to thank the Royal Society Open Science journal for accepting our manuscript, and the editors for their time and constructive comment to our manuscript. We have implemented the last suggestions from the editors into the manuscript and would like to submit the latest version of the manuscript.

Yours sincerely
On behalf of the co-authors
Signe Dille Løvmo

Comments from the reviewers:

The authors have satisfactorily addressed previous reviewer concerns, and this manuscript is acceptable for publication after some minor revisions. I have a few additional comments that might help to improve the clarity of the figures:

Figure 2: The colors and symbols are redundant. I would suggest to use the same color for $p < 0.05$ and $p < 0.01$ irrespectively if genes are up or down regulated. Also, please avoid red and green in the same graph to make sure readers with color blindness can interpret the results.

Figure 4 & 5: If possible, avoid the red-green gradient (see comment above).

Reply: we thank the reviewer for pointing this out, and have now given the same color to $p < 0.05$ and $p < 0.01$ in figure 2. We have also updated the colors in figure 2, 4 and 5 to be more suitable for readers with colorblindness.

Reviewer: 2
Comments to the Author(s)

The manuscript should be once more very carefully proofread for proper wording, grammar and punctuation.

Examples:

Line 147. Missing a period at the end of the sentence. Also should the number of that reference be between brackets?

Line 152: Sections we stained with.... Should be Sections were stained....

Line 179: change respectfully, with respectively.

Reply: We thank the reviewer for pointing this out and have taken time to carefully proofread the whole manuscript.